# Lineage commitment of embryonic cells involves MEK1-dependent clearance of pluripotency regulator Ventx2

Pierluigi Scerbo, Leslie Marchal, Laurent Kodjabachian*

Institut de Biologie du Développement de Marseille, Aix Marseille Univ, CNRS, Marseille, France

**Abstract** During early embryogenesis, cells must exit pluripotency and commit to multiple lineages in all germ-layers. How this transition is operated in vivo is poorly understood. Here, we report that MEK1 and the Nanog-related transcription factor Ventx2 coordinate this transition. MEK1 was required to make *Xenopus* pluripotent cells competent to respond to all cell fate inducers tested. Importantly, MEK1 activity was necessary to clear the pluripotency protein Ventx2 at the onset of gastrulation. Thus, concomitant MEK1 and Ventx2 knockdown restored the competence of embryonic cells to differentiate. Strikingly, MEK1 appeared to control the asymmetric inheritance of Ventx2 protein following cell division. Consistently, when Ventx2 lacked a functional PEST-destruction motif, it was stabilized, displayed symmetric distribution during cell division and could efficiently maintain pluripotency gene expression over time. We suggest that asymmetric clearance of pluripotency regulators may represent an important mechanism to ensure the progressive assembly of primitive embryonic tissues.

## Introduction

How embryonic pluripotent cells can maintain an uncommitted state as well as an unrestricted potential for multi-lineage commitment is a key and unresolved question in developmental and stem cell biology. Therefore, studying in vivo the links between factors that oppositely regulate pluripotency should help to better understand the transitory nature of this state during embryogenesis and its resumption in several human diseases. In mammals, epiblast cells of the developing blastula embryo appear to transit through a series of pluripotent states until gastrulation, which signs the global extinction of pluripotency and the rise of cell competence to somatically commit (*Hackett and Surani, 2014*; *Boroviak and Nichols, 2014*). In amphibians, cells of the blastula animal hemisphere are somatically pluripotent, and their broad potential is globally lost at gastrulation (*Snape et al., 1987*). Studies in vivo have been useful to characterize the core regulatory network of pluripotency, and to reveal its degree of conservation and plasticity during vertebrate evolution (*Boroviak and Nichols, 2014*; *Hackett and Surani, 2014*; *Morrison and Brickman, 2006*; *Scerbo et al., 2012*; *Buitrago-Delgado et al., 2015*; *Boroviak et al., 2015*). In vertebrates, the pluripotency regulatory network is centered on the Pou-V class of transcription factors (also referred as Oct4). Pou-V members *Pou5f3* and *Pou5f1* share functional homology in regulating the uncommitted state of progenitor cells (*Livigni et al., 2013*) and in reprogramming somatic cells to induced-Pluripotent Stem Cells (iPSCs) (*Tapia et al., 2012*; *Takahashi and Yamanaka, 2006*). Nanog has been discovered as a key component of pluripotency networks in both mouse embryonic stem cells (mESC) and pre-implantation epiblast (*Hackett and Surani, 2014*; *Boroviak and Nichols, 2014*). Nonetheless, phylogenetic, biochemical and functional analyses suggest that the role of Nanog in pluripotency is not conserved in all vertebrates, as the *Nanog* gene is absent in the *Xenopus* genus

*For correspondence: laurent. kodjabachian@univ-amu.fr

Competing interests: The authors declare that no competing interests exist.

(*Scerbo et al., 2014*) and teleostean *Nanog* does not support pluripotency during development (*Camp et al., 2009*; *Scerbo et al., 2014*). Recent analyses on *Xenopus* and zebrafish embryos suggest that Ventx transcription factors, belonging to the same NK family as Nanog (*Scerbo et al., 2014*), act as guardians of pluripotency during embryogenesis (*Scerbo et al., 2012*; *Zhao et al., 2013*). Ventx factors integrate the pluripotency network by coordinating and maintaining the activity of Pou-V factors (*Scerbo et al., 2012*; *Zhao et al., 2013*; *Cao et al., 2004*), and by regulating cell response to TGF-β/Smad pathways (*Zhu and Kirschner, 2002*; *Cao et al., 2004*). However, how the pluripotency network evolves to authorize the expression of lineage-specific genes in lower vertebrates is poorly understood. Pluripotency is maintained by a complex gene regulatory network associated with a specific epigenetic state both in vitro and in vivo (*Boroviak and Nichols, 2014*). Several studies revealed the importance of transcriptional and epigenetic silencing of pluripotency-related genes during the process of cell commitment, which ultimately allows the transcriptional activation of lineage-specific genes (*Hackett and Surani, 2014*). Based on in vitro studies, the regulation of pluripotency factor stability and degradation (*Kim et al., 2014*; *Spelat et al., 2012*), as well as the asymmetric distribution of cytoplasmic and membrane-bound determinants during cell division (*Habib et al., 2013*), are also expected to significantly contribute to pluripotency destabilization. However, whether such mechanisms are important during vertebrate embryogenesis remains to be addressed. Studies in mammalian embryos have suggested that MEK (*Map2k1*), an upstream component of the mitogen activated protein kinase (MAPK) pathway, could represent an intrinsic determinant of the ephemeral and transitory nature of the pluripotency state in vivo (*Boroviak et al., 2015*; *Boroviak and Nichols, 2014*). Further support to this idea comes from the reported stabilization in a pluripotent state of mammalian ESCs in vitro by media that include inhibitors of MEK1 activity (*Boroviak and Nichols, 2014*; *Hackett and Surani, 2014*; *Theunissen et al., 2014*). MEK1 activity can negatively regulate both the expression in vivo (*Boroviak et al., 2015*; *Boroviak and Nichols, 2014*) and the stability in vitro of Nanog and Pou5f1 proteins (*Kim et al., 2014*; *Spelat et al., 2012*). The *Xenopus* embryo also represents an attractive model to address how MEK1 controls pluripotency exit in vivo, as the importance of the MAPK pathway for the competence of embryonic cells to differentiate has long been known (*LaBonne et al., 1995*). Activation of the MAPK ERK1, resulting from phosphorylation by MEK1, is known to occur at early blastula stages, primarily in the pluripotent cells of the animal and marginal zones (*Curran and Grainger, 2000*). Multiple studies revealed that FGF-mediated ERK1 activation is necessary for the competence of animal blastula cells to respond to mesoderm and neural inducers (*Cornell and Kimelman, 1994*; *Delaune et al., 2005*). However, the existence of a direct link between the MAPK pathway and pluripotency during *Xenopus* development has never been tested. In this study, we reveal that MEK1 is required for embryonic cell competence to respond to differentiation cues, acting against the expression, distribution and stability of the pluripotency regulator Ventx2.

## Results

### MEK1 is required for cell competence to differentiate

To examine the role of MEK1 in *Xenopus* embryos, we depleted it through injection of morpholino antisense oligonucleotides (MOs). We designed two independent MOs, in the 5′UTR (Mk-MO), and at the ATG (Mk-MO ATG), which both inhibited MEK1 translation and antagonized development (*Figure 1A–D* and *Figure 1—figure supplement 1–1A*). As Mk-MO proved more efficient, we used it for most experiments in this study, unless stated otherwise. Mk-MO did not up-regulate *p53* expression (*Figure 1—figure supplement 1–1B*), unlike the non-specific response triggered by some MOs in zebrafish embryos (*Robu et al., 2007*). Importantly, a wild-type form of MEK1 from hamster efficiently rescued mesoderm and neural specification, as well as early morphogenesis, in Mk morphant embryos, indicating that MEK1 knockdown was specific (*Figure 1D* and *Figure 1—figure supplement 1–1C and D*). We found that MEK1 activity was required for the expression of multiple neural and non-neural ectoderm, as well as mesoderm markers (*Figure 1—figure supplement 1–2A and B*). In contrast, MEK1 activity was found to be dispensable for the expression of endoderm markers and of immediate-early targets of the BMP and Nodal pathways (*Figure 1—figure supplement 1–2C and D*). Interestingly, the expression of the epidermal markers *dlx3*, *gata2* and *xk81a1* was reduced in MEK1 morphants, despite the maintained expression of the epidermal inducer

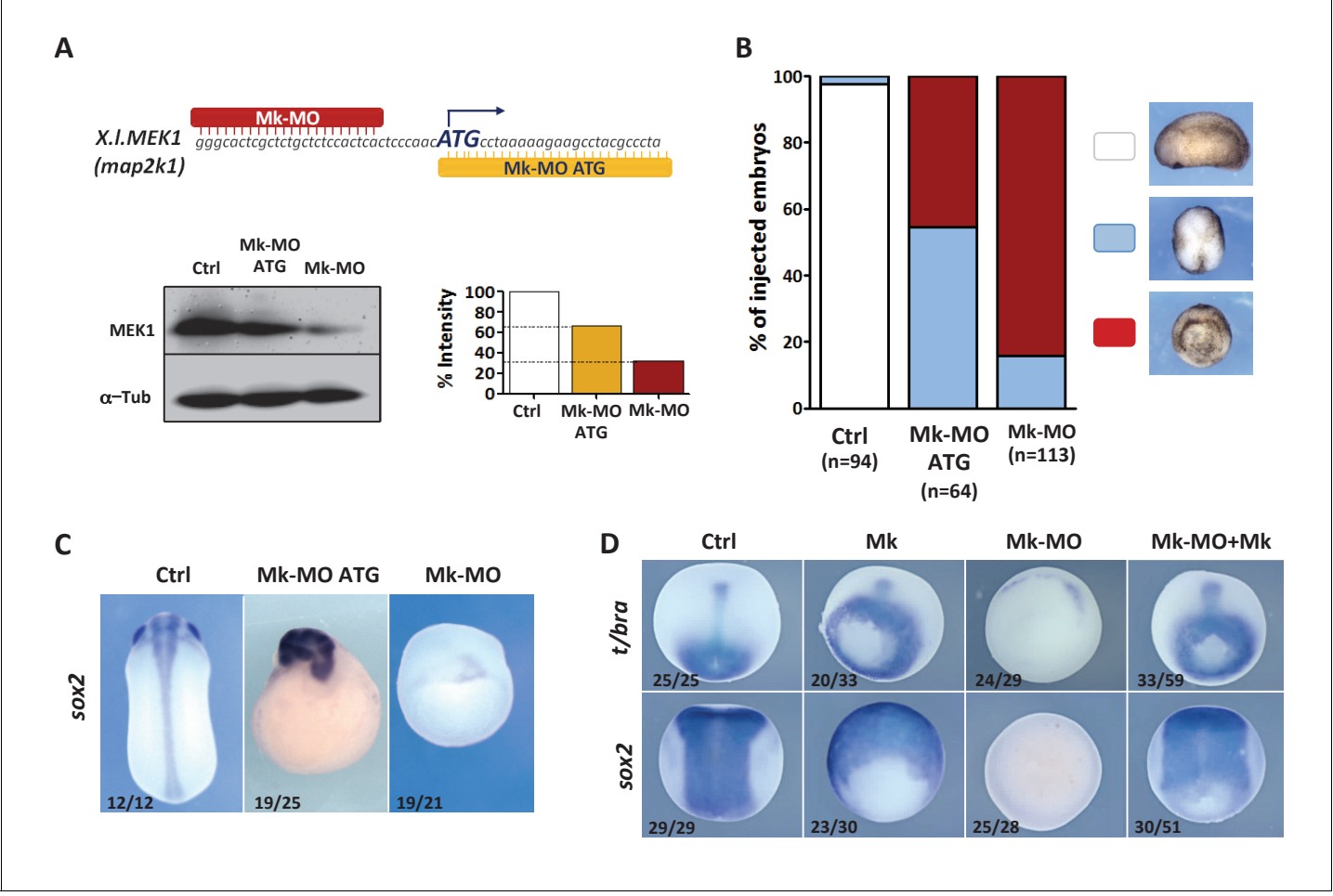

**Figure 1.** MEK1 depletion impairs embryonic development. (**A**) Mk-MO and Mk-MO ATG were designed to target MEK1 translation. Western blot analysis of blastula stage nine embryos injected with 25 ng per blastomere of either MO at the 4 cell stage revealed reduced MEK1 translation. α-Tubulin was used as a loading control. Control embryos were uninjected. The histogram shows the normalized intensity of MEK1 signals relative to control. (**B**) Embryos were injected as in (**A**) and morphology was analyzed at tailbud stage. (**C**) Embryos injected as in (**A**) were stained with *Sox2* probe to highlight defective axis formation and neural tissue differentiation. (**D**) Embryos were injected at the 2 cell stage with 25 ng Mk-MO per cell, and at the 4- cell stage with 400 pg of mammalian MEK1 (Mk) RNA per cell and processed for WISH analysis at late gastrula stage 13 with *t/bra* probe to highlight the mesoderm (dorso-vegetal view) and with *sox2* to highlight the neurectoderm (dorsal view). In C and D, the number of embryos exemplified by the photograph over the total number of embryos analyzed is indicated.

The following source data and figure supplements are available for figure 1:

**Figure supplement 1.** MEK1 depletion by morpholinos.

**Figure supplement 1—source data 1.** Values of blastopore closure ratios.

**Figure supplement 2.** Gene expression analysis of MEK1-depleted gastrula embryos.

*bmp4*. These results suggested that MEK1 was broadly required for multi-lineage commitment of pluripotent cells of the animal hemisphere. To further test this possibility, we examined the differentiation potential of embryonic cells depleted of MEK1 in response to exogenous inducers. Consistent with our hypothesis, recombinant BMP4, NODAL, and NOGGIN proteins induced efficient expression of epidermal, mesodermal and neural markers, respectively, in wild-type but not in MEK1-depleted cells (*Figure 2A*). Likewise, BMP signaling inhibition, obtained by injection of dominant-negative Smad5 (*Marchal et al., 2009*), caused neural induction in wild-type embryos, but not in

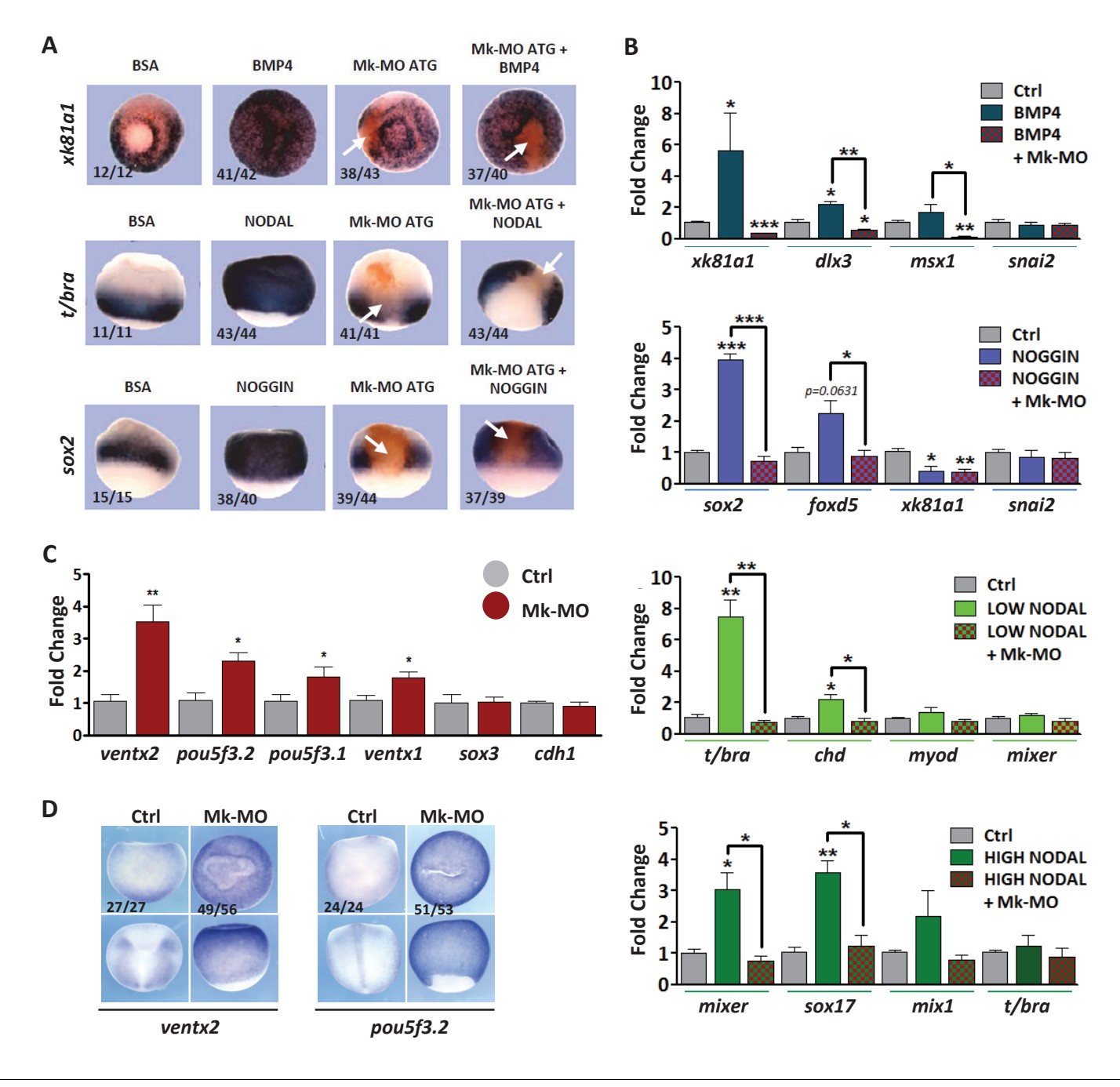

**Figure 2.** MEK1 depletion affects cell competence to exit pluripotency and enter into differentiation. (**A**) Sixteen-cell embryos were injected in one animal blastomere with 25 ng Mk-MO ATG and 2.5 ng FLDX. Next, these embryos were injected at blastula stage 8.5 with recombinant BMP4 (2 ng), NODAL (10 ng), or NOGGIN (36 ng) proteins into the blastocoele, collected at early gastrula stage 10.5, and processed for WISH with *xk81a1* (epidermis, animal view), *t/bra* (mesoderm, lateral view), and *sox2* (neural tissue, dorsal view). FLDX (orange staining) was used to trace Mk-MO injected cells (white arrows). (**B**) Four-cell embryos were injected with 25 ng Mk-MO per blastomere, animal caps were explanted at blastula stage 8.5 and cultured in the presence of 20 ng/ml BMP4, 100 ng/ml NOGGIN, 20 ng/ml NODAL (low), or 200 ng/ml NODAL (high) until late gastrula stage 13, and processed for RT-qPCR. (**C**) Four-cell embryos were injected with 25 ng Mk-MO per blastomere, collected at stage 10.5 and processed for RT-qPCR. (**D**) Embryos injected as in (**C**) were processed for WISH analysis at late gastrula stage 13 with *pou5f3.2* (*oct25*) and *ventx2* probes. a: animal view; v: ventral view; l: lateral view; d: dorsal view. For all qPCR graphs, error bars represent s.e.m. values of four independent experiments with two technical duplicates. For statistical analyses, samples were compared with the respective control by Unpaired Student's t-test. *$p < 0.05$, **$p < 0.005$. ***$p < 0.0001$. In A and D, the number of embryos exemplified by the photograph over the total number of embryos analyzed is indicated.

*Figure 2 continued on next page*

*Figure 2 continued*

The following figure supplements are available for figure 2:

**Figure supplement 1.** Neural induction in vivo depends on MEK1 activity.

**Figure supplement 2.** MEK1 is required to inhibit the expression of the pluripotency genes *pou5f3.2* and *ventx2*.

Mk-MO ATG morphants (*Figure 2—figure supplement 2–1*). To directly assess the importance of MEK1 in the competence of pluripotent cells to respond to inducers, we exposed explanted animal ectoderm to the soluble factors described above. Whereas multiple epidermal, neural, mesodermal and endodermal markers were induced by BMP4, NOGGIN, low and high doses of NODAL, respectively, these responses were all abolished in the presence of Mk-MO (*Figure 2B*). The impaired ability of MEK1-deficient cells to engage into differentiation could be linked to mis-regulation of pluripotency genes. RT-qPCR analysis on early gastrula embryos revealed that the expression of *pou5f3.1* (*oct91*), *pou5f3.2* (*oct25*), *ventx1* and *ventx2* was significantly up-regulated in MEK1-depleted embryos, whereas no detectable effect was measured on *sox3* and *cdh1* (E-cadherin) (*Figure 2B*)(*Figure 2C*). *ventx2* and *pou5f3.2* are initially expressed throughout the pluripotent animal hemisphere and are restricted to non-neural ectoderm and floor plate at late gastrulation, respectively, reflecting the progressive engagement of embryonic cells into differentiation. In contrast, these two genes remained ubiquitously expressed in MEK1 morphants at late gastrula stage (*Figure 2D*). Likewise, targeted injection of Mk-MO in dorsal or ventral ectoderm confirmed that *pou5f3.2* and *ventx2* expression failed to be silenced, even when morphogenesis was not altered (*Figure 2—figure supplement 2–2A*). Strikingly, the maintenance of *pou5f3.2* expression in MEK1 morphants was still visible at mid-neurula stage (*Figure 2—figure supplement 2–2B*). Furthermore, gene expression analysis of MEK1-depleted animal ectoderm explants at the end of gastrulation revealed a significantly higher expression of *pou5f3.2* and *ventx2* with a concomitant reduction of the lineage-restricted markers *xk81a1*, *itln1* and *α-tub* (*Figure 2—figure supplement 2–2C*). Taken together, these data suggested that embryonic cells need MEK1 activity to exit pluripotency and engage into lineage-specific programs.

## MEK1 regulates the sub-cellular distribution and clearance of Ventx2

The above results raised the possibility that an elevated activity of pluripotency factors may promote resistance to differentiation. In *Xenopus*, Ventx2 is involved in the active repression of precocious differentiation and helps to maintain pluripotency (*Scerbo et al., 2012*). Consistently, it was shown that Ventx2 undergoes ubiquitin-mediated degradation in the early gastrula (*Zhu and Kirschner, 2002*), which coincides with the global loss of pluripotency (*Snape et al., 1987*). Ventx2 proteolysis involves a PEST-destruction motif in its N-terminus, which is regulated by phosphorylation by uncharacterized signaling pathways (*Zhu and Kirschner, 2002*). Noteworthy, this PEST domain is conserved in the Ventx family (*Supplementary file 1*)(*Supplementary file 4*), and one of the two functional serines in the PEST motif of *Xenopus* Ventx2 is a predicted target of MAPK (*Figure 3—figure supplement 1A and B*). Thus, we addressed whether MEK1 may participate in the control of Ventx2 stability. Western blot analysis of embryos injected with RNA encoding Myc-tagged Ventx2 confirmed the reported degradation after the onset of gastrulation (*Figure 3—figure supplement 1C*). In contrast, in MEK1 morphants, Ventx2-Myc remained detectable up until mid-gastrula stage (*Figure 3—figure supplement 1C*). Next, we analyzed Ventx2-Myc distribution on sectioned blastula and gastrula embryos by immunofluorescence confocal microscopy. We found that MEK1 knockdown dramatically increased the number of Ventx2-Myc positive cells at both stages, when compared to control (*Figure 3A*). Careful inspection of blastula cells dividing along the animal-vegetal (apical-basal) axis of the tissue revealed that Ventx2-Myc was asymmetrically distributed in daughter cells in control but not in MEK1 morphant embryos (*Figure 3B and C*). In addition, a significant fraction of MEK1-depleted cells displayed Ventx2-Myc signal in the membrane cortex at both stages analyzed (*Figure 3A and B*). At blastula stage, Ventx2-Myc membrane association was polarized with respect to the plane of the tissue, with a clear basal enrichment (*Figure 3A and B*). No attempt was made at characterizing further the cortical localization of Ventx2. Altogether, our data reveal that MEK1 is

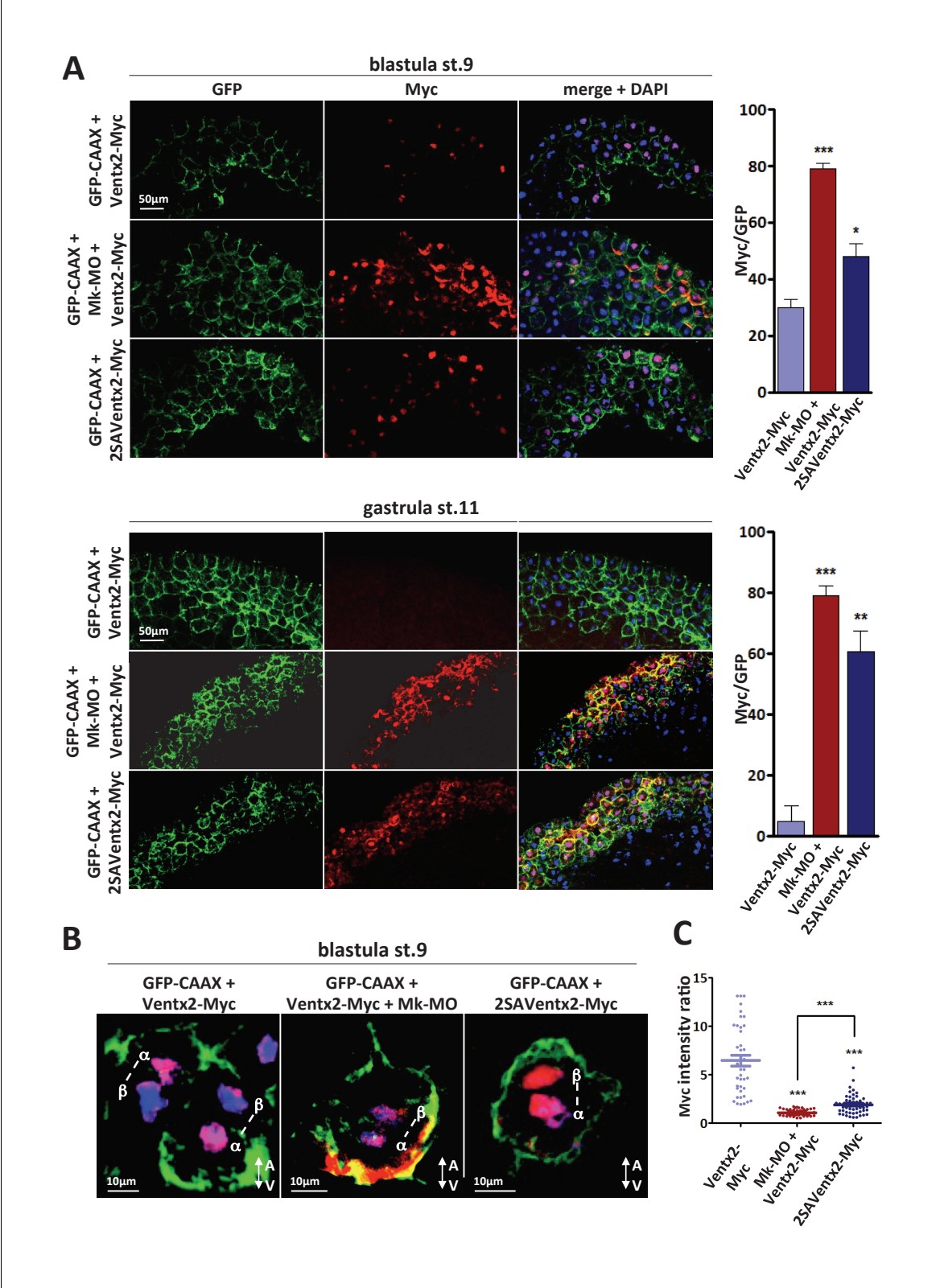

**Figure 3.** MEK1 is required for Ventx2 clearance and asymmetric distribution during cell division. (A,B) Four-cell embryos were injected in each cell with 50 pg GFP-CAAX, 50 pg Ventx2-Myc, 50 pg 2SAVentx2-Myc RNAs, and 25 ng Mk-MO, as indicated. Embryos were fixed at blastula stage 9, or gastrula stage 11, cryosectioned and processed for anti-Myc (red), and anti-GFP (green) immunostaining, and DNA was stained with DAPI (blue). Graphs show the percentage of Myc positive nuclei (DAPI positive) over the total number of injected cells (GFP positive) from four independent experiments. (B) 3D

*Figure 3 continued on next page*

*Figure 3 continued*

reconstruction of confocal slices of mitotic Myc positive nuclei labeled by DAPI from stage nine sectioned embryos. Sister mitotic chromosomes are referred to as α (more intense Myc staining), and β (less intense Myc staining). The A-V arrows indicate the animal-vegetal axis. Note the asymmetric cortical Ventx2-Myc signal in the MEK1 morphant cell. (C). The graph shows the ratios of Myc signal intensity between α and β sister nuclei.

The following source data and figure supplement are available for figure 3:

**Source data 1.** Myc signal intensity ratios between daughter nuclei.
**Figure supplement 1.** Ventx2 degradation and asymmetric distribution require MEK1 activity.

required for Ventx2 asymmetric distribution during blastula cell division, and participates in developmentally regulated clearance of this protein in pluripotent embryonic cells.

## The PEST destruction motif is required for destabilization and asymmetric distribution of Ventx2

The above results raised the possibility that destabilization of Ventx2 in response to MEK1 is a key step for the transition between pluripotent and committed states of embryonic cells. To further evaluate this possibility, we injected a mutant form of Ventx2 lacking the functional PEST destruction motif. This stable form of Ventx2 was obtained through the substitution of two key serine residues (140 and 144) with alanine (2SAVentx2-Myc) and was demonstrated to be more efficient than the native form of Ventx2 in counteracting mesendoderm induction (*Zhu and Kirschner, 2002*). As predicted, 2SAVentx2-Myc was detectable in the nuclei of control cells, similar to Ventx2-Myc in MEK1 morphant cells (*Figure 3A*). Moreover, no marked asymmetric distribution of 2SAVentx2-Myc in daughter nuclei of dividing blastula cells was observed, suggesting that phosphorylation of the PEST motif is necessary to control Ventx2 asymmetric distribution (*Figure 3B and C*). Similarly to Ventx2-Myc in MEK-1 deficient cells, 2SAVentx2-Myc was detectable in the cortex of injected cells, particularly at gastrula stage (*Figure 3A*). Next, we combined anti-Myc and γ-tubulin staining to mark the centrosomes, so as to be able to distinguish mitotic phases (*Figure 3—figure supplement 1D*). Ventx2-Myc localized into the nuclei of blastula cells in prophase and MEK1 depletion did not affect this localization. Similarly, 2SAVentx2-Myc co-localized with chromosomes during prophase. Ventx2 is a phosphomitotic protein (*Stukenberg et al., 1997*), and it has been proposed that phosphorylation is required for disengagement of transcription factors from chromosomes in metaphase (*Stukenberg et al., 1997*). Accordingly, Ventx2-Myc was no longer associated with metaphasic chromosomes but rather with mitotic spindles. Interestingly, this change in distribution of Ventx2-Myc was independent from MEK1 activity or from the PEST destruction motif. Upon engagement into cytokinesis, the asymmetry of Ventx2-Myc in daughter nuclei became apparent. Quantification of signals intensity revealed that in control condition, one daughter nucleus contained on average 6.454 fold more Ventx2-Myc than the other daughter nucleus (s.e.m. ±0.5479), whereas this difference was significantly dampened in MEK1-deficient cells (mean ratios = 1.092; s.e.m. ±0.0447) and in 2SAVentx2-Myc injected cells (mean ratios = 1.910; s.e.m. ±0.1319)(*Figure 3C*). Thus, the stabilization of exogenous Ventx2 through either MEK1 depletion or PEST motif mutation led to its symmetric distribution upon cell division.

## Ventx2 inhibition by MEK1 is required for embryonic cell commitment

Our data indicate that Ventx2 expression, distribution and stability depend on MEK1 activity in vivo, raising the possibility of a causal relationship between the refractory behaviour of MEK1-depleted cells to differentiation and the activity of Ventx2. Consistent with this idea, 2SAVentx2-Myc, but not native Ventx2-Myc, caused a markedly high and ectopic expression of *pou5f3.2* in late gastrula embryos, which was visible up until tailbud stage (*Figure 4A* and *Figure 4—figure supplement 1A*). Furthermore, we found that 2SAVentx2-Myc but not Ventx2-Myc could counteract the conversion of ectoderm to mesoderm in response to a constitutively active form of MEK1 (*Figure 4B*). This result suggests that 2SAVentx2-Myc has become resistant to degradation induced by MEK1 and can efficiently maintain pluripotency and antagonize commitment in the presence of active MEK1. To test the functional importance of the antagonism between MEK1 and Ventx2 in the transition from

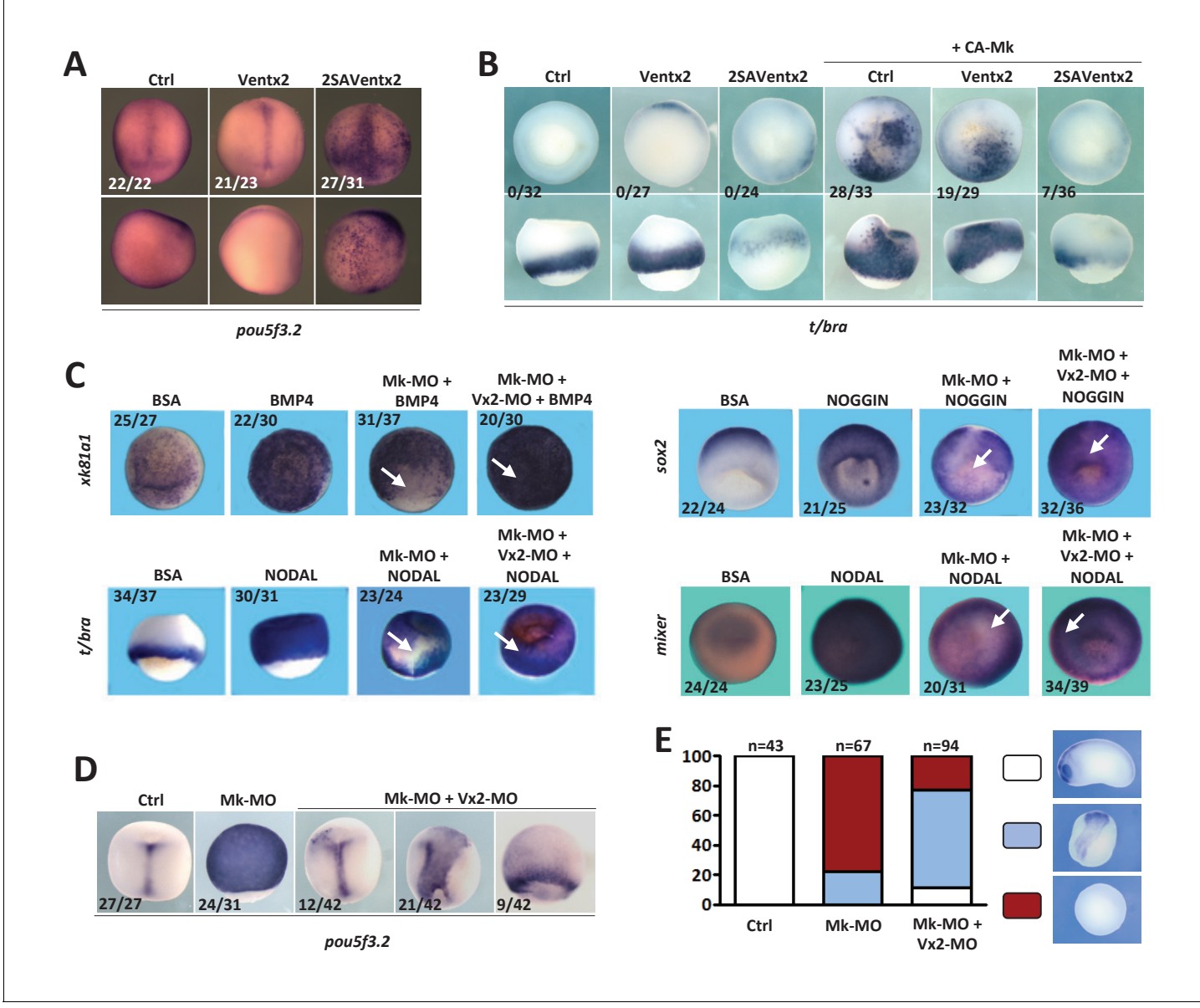

**Figure 4.** Ventx2 knockdown rescues the competence of MEK1-deficient cells to differentiate. (A) 4 cell embryos were injected in each cell with 50 pg Ventx2-Myc or 50 pg 2SAVentx2-Myc RNAs, and processed for WISH analysis at late gastrula stage 13 with *pou5f3.2* probe. Stabilized 2SAVentx2 maintains ectopic *pou5f3.2* expression. Top row dorsal view, bottom row lateral view. (B) Embryos injected as in (A), with or without 500 pg CA-Mk RNA per blastomere, were processed for WISH analysis at early gastrula stage 10.5 with *t/bra* probe. The number of embryos with ectopic *t/bra* expression is indicated in each condition. Stabilized 2SAVentx2 counteracts CA-Mk activity. Top row animal view, bottom row lateral view. (C) Sixteen-cell embryos were injected in one animal blastomere with 50 pg GFP-CAAX RNA, 25 ng Mk-MO and 7.5 ng Vx2-MO, as indicated. Next, these embryos were injected at blastula stage 8.5 with recombinant BMP4 (2 ng for *xk81a1* induction, animal view), NODAL (5 ng for *t/bra* induction, lateral view; 50 ng for *mixer* (endoderm) induction, animal view), or NOGGIN (36 ng for *sox2* induction, animal view) proteins into the blastocoel, collected at early gastrula stage 10.5, and processed for WISH. Mk-MO injected domains are indicated by white arrows. (D) Four-cell embryos were injected in each blastomere with 25 ng Mk-MO alone or with 7.5 ng Vx2-MO, collected at late gastrula stage 13 and processed for WISH with *pou5f3.2* probe. (E) Embryos injected as in (D) were collected at tailbud stage 25, processed for WISH with *sox2* probe, and scored. Ventx2 knockdown partially restores development of MEK1-deficient embryos. MEK1/Ventx2 double knockdown rescue assays were repeated five times. In A and C, the number of embryos exemplified by the photograph over the total number of embryos analyzed is indicated.

The following figure supplement is available for figure 4:

**Figure supplement 1.** Ventx2 knockdown restores germ-layer formation in MEK1-deficient embryos.

pluripotent to committed states, we combined Mk-MO with Ventx2 MO (*Scerbo et al., 2012*). RT-qPCR and in situ hybridization analyses revealed that Ventx2 MO injection caused the down-regulation of *ventx1* and *ventx3* expression, suggesting that a large part of Ventx activity is missing in such embryos (*Figure 4—figure supplement 1B and C*). As predicted, MEK1-Ventx2 double morphant cells regained the competence to respond to exogenous inducers and thus to express epidermal, neural, mesodermal and endodermal markers in response to BMP4, NOGGIN, low and high doses of NODAL, respectively (*Figure 4C*). Furthermore, the expression of the lineage-restricted markers *foxd5*, *gsc*, *t/bra*, and *xk81a1* was rescued in MEK1-Ventx2 double morphant embryos (*Figure 4—figure supplement 1D*). Conversely, the sustained expression of *pou5f3.2* caused by MEK1 depletion was counteracted by concomitant Ventx2 knockdown (*Figure 4D*). Finally, morphogenesis and axis formation was also partially restored in MEK1-Ventx2 double morphant embryos (*Figure 4E*). Altogether, these data indicate that MEK1 and Ventx2 functionally interact during the transition of pluripotent cells from refractory to responsive states.

## Discussion

The findings presented here reveal a mechanism for the control of embryonic cell competence to differentiate, which is linked to the spatio-temporal stability of pluripotency factors in vivo. Specifically, MEK1 activity counteracts Ventx2 activity, both at transcriptional and post-translational levels, which ensures the transition from refractory to responsive states of embryonic cells. In pluripotent blastula cells, MEK1 is necessary for asymmetric distribution of Ventx2 during cell division, generating Ventx2 positive and negative daughter nuclei. By promoting early heterogeneous distribution together with developmentally regulated nuclear clearance of Ventx2, we propose that MEK1 is a fundamental cue for pluripotency state extinction in vivo. To our knowledge, this is the first reported regulatory mechanism that correlates the asymmetric distribution and stability of a transcription factor with cell pluripotency in vertebrate embryos. Although we used exogenous Ventx2 tagged proteins to reveal this mechanism, the rescue of cell differentiation in double MEK1/Ventx2 morphants suggests that endogenous Ventx2 is under the same control. Our data are consistent with a recent finding, whereby the Wnt/β-catenin pathway orients asymmetric cell division of mESCs and generates unequal distribution of pluripotency factors, which impacts on cell potency in vitro (*Habib et al., 2013*).

How does MEK1 regulate Ventx2 asymmetry and stability in *Xenopus* embryonic cells? Ventx2 phosphorylation in the PEST motif was shown to be required for ubiquitination and degradation (*Zhu and Kirschner, 2002*). Thus, our observation that mutation of the Ventx2 PEST motif prevents unequal distribution in daughter nuclei suggests that the asymmetric distribution of Ventx2 during mitosis and proteasome activity may be related. In the simplest scenario, MEK1 may act directly or through MAPK to phosphorylate the Ventx2 PEST motif and promote Ventx2 proteolysis. Supporting this hypothesis, the MAPK ERK1 is a known regulator of pluripotency factor stability in vitro (*Spelat et al., 2012*; *Kim et al., 2014*), and MAPK is predicted to phosphorylate one of the key serine residues in the Ventx2 PEST domain. We note, however, that MEK1 depletion caused stronger responses than 2SAVentx2 injection, suggesting that MEK1 regulates pluripotency through additional residues in Ventx2 and/or through additional effectors. Interestingly, the BMP signal mediator Smad1 was also found to be asymmetrically distributed during somatic cell division, when marked for degradation by MAPK (*Fuentealba et al., 2008*). There, phosphorylation by MAPK triggers a subsequent phosphorylation event by GSK3, followed by polyubiquitinylation and proteosomal degradation of activated Smad1, leading to BMP signal termination. In contrast, we did not find evidence of GSK3 involvement in the repression of pluripotency genes and in Ventx2 degradation (*Supplementary file 1*), consistent with previous data (*Zhu and Kirschner, 2002*). Thus, MEK1 may trigger important proteolytic events in early embryos, independently of GSK3. Although it is tempting to think that MEK1 activity itself may be polarized during division of pluripotent embryonic cells, evidence for such mechanism is lacking in the literature. Alternatively, it is possible that MEK1 controls downstream regulators endowed with asymmetric distribution or activity. For instance, ERK1 was shown to antagonize the polarized PAR-1 kinase during asymmetric cell divisions of the early *C. elegans* embryo (*Spilker et al., 2009*). Also related to this idea, asymmetric proteasome segregation was shown to control polarized degradation of the phosphorylated transcription factor T-bet during T lymphocyte division (*Chang et al., 2011*). Finally, it remains possible that MEK1 activity would be

antagonized by one or several phosphatases that would protect Ventx2 from degradation, and would somehow contribute to its unequal inheritance in daughter nuclei. Such a balance between the PAR-1 kinase and PP2A phosphatase was shown to control the state of PAR-3 phosphorylation and thus the polarity of dividing embryonic neuroblasts in *Drosophila* (*Krahn et al., 2009*). Future work should address the precise mechanism of action of MEK1 upon Ventx2 during division of *Xenopus* pluripotent cells, a question of high relevance to stem cell biology.

Unexpectedly, we observed that MEK1 depletion caused Ventx2 asymmetric localization in the basal membrane cortex of blastula cells. This localization was not apparent in control cells, suggesting that it does not simply reflect the over-abundance of Ventx2 protein due to synthetic RNA injection. Rather, this suggests that MEK1 actively prevents membrane association of Ventx2. As stabilized 2SAVentx2 also localized at membranes, this territory may represent a storage compartment for Ventx2 protein. Thus, it will be important to evaluate whether the endogenous Ventx2 protein also displays this unexpected localization in normal or MEK1 morphant embryos. The asymmetric localization of Ventx2 at the basal membrane cortex of MEK1 depleted cells suggests possible links between MEK1 and polarity effectors such as PAR-1 and aPKC (*Ossipova et al., 2009*; *Chalmers et al., 2003*). In relevance to our observations, it was reported that Pou5f1 and Pou5f3 proteins localize at the cell membrane both in mESCs and in *Xenopus* animal pole cells, where they form a complex with E-cadherin and $\beta$-catenin (*Livigni et al., 2013*; *Faunes et al., 2013*). Since Ventx2 can physically interact with Pou5f3 proteins (*Cao et al., 2004*), we speculate that MEK1 may be required to destabilize the interaction between Ventx2 and Pou5f3, not only in the nucleus but also at the membrane, further enhancing the competence of embryonic cells to exit pluripotency.

Ventx2 is a *bona fide* marker of pluripotency during *Xenopus* embryogenesis (*Scerbo et al., 2012*; *Buitrago-Delgado et al., 2015*), and the above data indicate that its activity must be inhibited for cells to engage into differentiation pathways. However, the link between *Ventx* genes and pluripotency in other vertebrates, particularly in mammals, has not been actively studied. This may reflect the absence of Ventx orthologs in the rodent genus, although a unique *VENTX* gene is present in human (*Supplementary file 3*). Phylogenetic and synteny analyses suggest that a *Ventx* gene appeared at the base of gnathostome evolution, and its prototypical genomic locus has not changed for 450 My (*Supplementary files 2* and *3*). Human *VENTX* is the ortholog of *Xenopus*, birds, sauropside and coelacanth *Ventx2* (*Supplementary files 2* and *4*). Interestingly, a recent study reported a 6-fold up-regulation of *VENTX* (higher than *NANOG*, *PRDM14*, *POU5F1* and *SOX2*) in naive hESCs compared to conventional hESCs (*Theunissen et al., 2014*). In this study, naive hESCs were obtained in the presence of inhibitors of five kinases, including MEK1, suggesting that repression of VENTX by MEK1 may be conserved in human pluripotent cells. Beyond such circumstantial evidence, the next question is whether VENTX is an important regulator of pluripotency in human, as it is in *Xenopus*. Initial support for this idea comes from the identification of VENTX in an unbiased functional screen as a positive regulator of *Pou5f1* expression in hESCs (*Chia et al., 2010*). Together with the results reported here and elsewhere (*Scerbo et al., 2012*; *Buitrago-Delgado et al., 2015*), such evidence certainly grants more detailed analysis regarding the role of VENTX in the human pluripotency network. As VENTX is absent in rodents, we propose that *Xenopus* represents an appropriate and powerful model to undertake comparative approaches with human and shed light on the control mechanisms of pluripotency in vivo and at single cell level.

## Materials and methods

### *Xenopus* general procedures and micro-injections

*Xenopus laevis* embryos were obtained from lab-bred adults (Nasco) by in vitro fertilization, de-jellied, injected and cultured in modified Barth's solution (MBS) as previously described (*Marchal et al., 2009*). Capped mRNAs for injection were synthesized with mMessage mMachine kits (Ambion, Austin, Texas). When necessary, lineage tracing was achieved through co-injection of Fluorescent Lysine DeXtran (FLDX) or membrane bound GFP-CAAX revealed by anti-FLDX or anti-GFP immune staining. The pCS2-Ventx2-Myc and pCS2-2SAVentx2-Myc (previously reported as *Xom* but now referred as *ventx2.2*, Xenbase-*Xenopus* Genome Initiative) plasmids were used as described (*Zhu and Kirschner, 2002*). The rabbit pCS2-CA-MEK1 plasmid was linearized with NcoI and mRNA synthesized with SP6 polymerase. To rescue Mk morphant embryos, the hamster pECE-

MEK1 plasmid (*Pagès et al., 1994*) was digested with XbaI and HindIII to isolate MEK1 ORF, which was subsequently cloned into the pSP64T vector. This new construct was linearized with BamHI and mRNA synthesized with SP6 polymerase. The dominant-negative GSK3 (dnGSK3) expression construct was used as previously described (*Puppo et al., 2011*). Mk-MO ATG (5'-TGGGCGTAGGCTTC TTTTTAGGCAT–3') and Mk-MO (5'-TGAGTGGAGAGCAGAGCGAGTGCCC–3') were purchased from GeneTools, LLC. The Vx2-MO was described in (*Sander et al., 2007*). BMP4 (R&D System; 314 BP), NODAL (R&D System; 1315-ND) and NOGGIN (R&D System; 334 NG) proteins were resuspended as recommended by manufacturers, and injected through the animal pole into the blastocoelic cavity of blastula stage embryos, or added in 1xMBS to culture animal cap explants. Throughout the study, each injection experiment was performed three or more times on different batches of embryos. Rescue experiments using Mk-MO and wild-type MEK1 were repeated five times, and blastopore closure quantification was performed on three independent experiments as previously described (*Martinez et al., 2015*). The number of embryos analyzed by condition ranged between 20 and 100. In total, over 700 MEK1 morphant embryos were analyzed with various methods and markers, with a very high penetrance of the reported phenotypes.

## Stainings

Whole-mount chromogenic in situ hybridization (WISH) was performed as previously described (*Marchal et al., 2009*), and photographs were taken on a Zeiss stereomicroscope equipped with a DS-L2 Nikon camera. Plasmids used to make antisense riboprobes are described in *Supplementary file 5*. FLDX was detected by incubation with alkaline phosphatase conjugated antifluorescein antibody (dilution 1/10,000; Roche). Sections were prepared and immune staining was performed as previously described (*Cibois et al., 2015*; *Castillo-Briceno and Kodjabachian, 2014*). Primary antibodies were as follows: anti-Myc (9E10; Santa Cruz Biotech, dilution 1/300 RRID:AB_ 627268), anti-GFP (GFP-1020; 2BScientific; dilution 1/1000 RRID:AB_10000240), anti-γ-Tubulin (ab16504; Abcam; dilution 1/1000 RRID:AB_443396), anti-phospho-MEK1 (9121; Cell Signaling Technology; dilution 1/400 RRID:AB_331648). Alexa Fluor secondary antibodies (Molecular Probes) were used at a dilution of 1:500. To stain DNA, DAPI (Invitrogen), at a final concentration of 10 μg/ml, was added to one of the final MABX washes and incubated for 3 min at room temperature. Stained sections were mounted with Fluoromount G (Fluoprobes) and allowed to dry before imaging on a Zeiss LSM780 confocal microscope. Images were acquired as eight bit/channel and with $1024 \times 1024$ pixel resolution, and processed with ImageJ (RRID:SCR_003070) for maximum intensity z-projection and/or merge of channels. Z-projections of green channel images were used to count GFP-positive injected cells. The percentage of Ventx2 positive nuclei was determined using a merge of Myc and GFP channels in order to consider only co-injected cells. For blastula stage 9 quantification of the percentage of Myc positive nuclei, a total of 251, 244 and 238 GFP-positive cells from Ventx2-Myc, Mk-MO+Ventx2-Myc, 2SAVentx2-Myc injected embryos was analyzed. For gastrula stage 11 quantification of the percentage of Myc positive nuclei, a total of 213, 238 and 207 GFP-positive cells from Ventx2-Myc, Mk-MO+Ventx2-Myc, 2SAVentx2-Myc injected embryos was analyzed. For statistical analyses, samples from Mk-MO+Ventx2-Myc and 2SAVentx2-Myc injected embryos were compared with samples from Ventx2-Myc injected embryos (as control) by Unpaired Student's t-test with Welch's correction (95% of confidence interval), and error bars represent s.e.m. values. To analyze dividing cells, fluorescence intensity levels of Myc-tagged Ventx2 proteins were measured using ImageJ (RRID:SCR_003070), from stacks of confocal images from 5 to 10 sections per independent experiment (at least four for each stage analyzed). For quantification of Myc-intensity ratios between $\alpha$ and $\beta$ daughter nuclei, stack-by-stack calculation of the ratios of fluorescence intensity of Ventx2-Myc (n = 42 stacks) alone or with Mk-MO (n = 51 stacks), or of 2SAVentx2-Myc (n = 56 stacks) from four dividing cells per case was performed. Non-parametric Mann-Whitney *U* test (95% confidence interval) was used to assess statistical differences among samples and error bars represent s.e.m. values. Statistical analysis was made using GraphPad Prism 6 (RRID:SCR_ 002798). For western blotting, embryos were snap-frozen and processed as described (*Luxardi et al., 2010*). Proteins were transferred to PVDF membranes (Bio-Rad) and analyzed by immunoblotting with appropriate primary antibodies: anti-Myc (9E10; Santa Cruz Biotech, dilution 1/ 100 RRID:AB_627268), anti-GFP (GFP-1020; 2BScientific; dilution 1/200 RRID:AB_10000240); anti-MEK1 (4A5; Cliniscience; dilution 1/1000 RRID:AB_2042302), anti-α-tubulin (DM1A; AbCam; dilution 1/1000 RRID:AB_2241126). HRP-conjugated were used as secondary antibodies (1/5000, Dako).

Immunoreactive bands were detected using the Immobilon ECL Kit (Merck Millipore) on a LAS-3000 imager (Fujifilm).

## Reverse transcriptase quantitative polymerase chain reaction (RT-qPCR)

Mk-MO injected and uninjected (controls) embryos were grown until gastrula stage 10.5, and then processed for RT-qPCR, as previously described (*Scerbo et al., 2012*; *Castillo-Briceno and Kodja-bachian, 2014*). Ten embryos per biological replicate were used. Animal pole explants from injected and uninjected embryos were taken at blastula stage 9, grown in 1X MBS until late gastrula stage 13, and processed for RT-qPCR. 15 animal pole explants per biological replicate were used. RT-qPCR Primers are listed in *Supplementary file 6*. Statistical analyses were done using GraphPad Prism 6 (RRID:SCR_002798)

## Acknowledgements

We thank Pr. Zhu and Kirschner for Ventx2 constructs. We thank Andrea Pasini and Vincent Bertrand for helpful comments. This work was supported by Centre National de la Recherche Scientifique (CNRS), Aix-Marseille Université, and by grants from Fondation pour la Recherche Médicale (DEQ20141231765), Fondation ARC (PJA 20141201815), and Institut National du Cancer (2012–108). Authors acknowledge France-BioImaging infrastructure funding 'Investissements d'Avenir' (ANR-10-INSB-04–01).

## Additional information

### Funding

| Funder | Grant reference number | Author |
| --- | --- | --- |
| Fondation pour la Recherche Médicale | DEQ20141231765 | Laurent Kodjabachian |
| Fondation ARC pour la Recherche sur le Cancer | PJA 20141201815 | Laurent Kodjabachian |
| Institut National Du Cancer | 2012-108 | Laurent Kodjabachian |

The funders had no role in study design, data collection and interpretation, or the decision to submit the work for publication.

### Author contributions

PS, Conceptualization, Formal analysis, Investigation, Writing—original draft; LM, Investigation; LK, Conceptualization, Supervision, Funding acquisition, Investigation, Writing—original draft, Project administration, Writing—review and editing

### Author ORCIDs

Laurent Kodjabachian, http://orcid.org/0000-0002-4000-612X

### Ethics

Animal experimentation: All the experiments were performed following the Directive 2010/63/EU of the European parliament and of the council of 22 September 2010 on the protection of animals used for scientific purposes. All animal experiments were approved by the "Direction départementale de la Protection des Populations, Pôle Alimentation, Santé Animale, Environnement, des Bouches du Rhône" (agreement number E 13-055-21).

## Additional files

### Supplementary files

• Supplementary file 1. GSK3 is not a negative regulator of the pluripotency gene network and is not required for MEK1-dependent Ventx2 clearance in vivo. (A) Four-cell embryos were injected with

300 pg DN-GSK3 RNA per blastomere. Embryos were collected at stage 25 and processed for WISH analysis with *sox2* probe. DN-GSK3 efficiently induced secondary body axes, indicating that the dose used was functional. (B) Embryos injected as in (A), were collected at stage 10.5 and processed for RT-qPCR. (C-D) Embryos injected as in (A) were processed for WISH analysis at early gastrula stage 10.5 with *gsc* (C, vegetal view) and *ventx2* (D, top: vegetal view, bottom: animal view) probes. (E) Embryos injected at the 8 cell stage with 300 pg DN-GSK3 RNA in one dorsal animal blastomere were processed for WISH analysis at late gastrula stage 13 with *pou5f3.2* and *ventx2* probes (anterior view). (F-G) Four-cell embryos were injected in each cell with 50 pg GFP-CAAX, 50 pg Ventx2-Myc, and 25 ng Mk-MO (F), or 50 pg GFP-CAAX, 50 pg Ventx2-Myc and 300 pg of DN-GSK3 (G) Animal caps were explanted at blastula stage nine and cultured until gastrula stage 11, fixed and processed for anti-Myc (red), and anti-GFP (green) immunostaining, and DNA was stained with DAPI (blue). Note that Ventx2-Myc is detectable only in MEK1 depleted caps. For the qPCR graph, error bars represent s.e.m. values of three independent experiments with two technical duplicates. For statistical analyses, samples were compared with the respective control by Unpaired Student's t-test. *p<0.05, **p<0.005. In C, D and E, the number of embryos exemplified by the photograph over the total number of embryos analyzed is indicated. In F and G scale bar is 20 µm.

• Supplementary file 2. Phylogenetic tree of *Ventx* deuterostome genes. Boxes indicate members with orthology relationship, like coelacanth *Ventx*, *Xenopus Ventx2* and human *VENTX* (blue arrows). Sequences were collected from ENSEMBL, JGI, A-STAR and NCBI public databases (see *Supplementary file 4*). Ventx homeodomain sequences were aligned using Jalview software (RRID: SCR_006459) and the phylogenetic tree was obtained by Neighbor Joining analysis of percentage identity.

• Supplementary file 3. The Evolutionary history of Ventx family genes. (A) Synteny of the *Ventx* genomic region in gnathostomes. Blue dotted boxes indicate species-specific gene duplication events. Note that a triplication event, giving rise to *Ventx1*, *Ventx2* and *Ventx3*, occurred in the last common ancestor of tetrapods. One or more *Ventx* paralogs was subsequently lost during squamata, archosaura and testudina evolution. Mammals lost both *Ventx1* and *Ventx3* paralogs and exclusively kept *Ventx2*. Mouse represents an extreme case with a total loss of *Ventx* genes. (B) Simplified tree of vertebrates, which displays typical situations regarding the number of *Ventx* genes in main evolutionary branches.

• Supplementary file 4. EMBOSS prediction of PEST destruction motifs in Ventx orthologs.

• Supplementary file 5. Probes used for WISH.

• Supplementary file 6. Primers used for RT-QPCR.

• Source data 1. Related to *Supplementary file 2*. VENTX homeodomain sequences.

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
