## [Decision Letter]

[Editors’ note: this article was originally rejected after discussions between the reviewers, but the authors were invited to resubmit after an appeal against the decision.]

Thank you for submitting your work entitled "Lineage commitment of embryonic cells involves MEK1-dependent clearance of pluripotency regulator Ventx2" for consideration by *eLife*. Your article has been reviewed by two peer reviewers, and the evaluation has been overseen by a Reviewing Editor and a Senior Editor.

Our decision has been reached after consultation between the reviewers. Based on these discussions and the individual reviews below, we regret to inform you that your work will not be considered further for publication in *eLife*.

Although we agree that your work is of potential interest, further experimentation is necessary to convince them of the validity of your conclusions. Among other studies, you would need to use animal caps instead of whole embryos, carry out a more comprehensive analysis to definitely rule out roles for other signaling pathways, perform other types of confirmatory experiments that don't rely on the use of morpholinos and address the problem of maternally-contributed MEK1 as a confounding variable.

Reviewer #1:

This manuscript uses the intact *Xenopus laevis* embryo as a model system to elucidate the molecular mechanisms involved in the regulation of pluripotency, focussing on MEK1 (*map2k1*) regulation of Ventx2.2 (encoded by one of a cluster of six related Ventx genes in *X. laevis*). Ventx proteins, absent in the mouse but present in human, appear to be functionally analogous to the transcription factor *Nanog* (absent in *X. laevis*).

The authors demonstrate that MEK1 activity acts to modulate exogenous (and presumably endogenous) Ventx2.2 protein stability; in response to morpholino-mediated down-regulation of MEK1, Ventx2.2 levels increase, the post-mitotic asymmetry in nuclear (exogenous) Ventx2.2 protein disappears, and cellular differentiation is suppressed. The cell differentiation phenotype is rescued in MEK1 morphant embryos by the co-injection of a translation-blocking anti-Ventx2 morpholino.

Experimental design question: While the authors recognize that *Xenopus* ectodermal explants (animal caps) are effectively pluripotent they have chosen to carry out their studies in whole embryos, which are subject to the complexities of inductive interactions and morphogenetic processes.

Perhaps they could explain their decision, and comment on whether they have examined differentiation in the simpler animal cap system.

Reviewer #2:

This work explores the role of MEK and the transcription factor Ventx2 (similar to mammalian *Nanog*) in regulating the transition from pluripotency to lineage commitment in *Xenopus*. The authors show that MEK knockdown results in a failure of mesoderm and ectoderm development, based on marker gene analysis. The authors further provide evidence that MEK acts by negatively regulating the mRNA levels and protein stability of Ventx2. They report that Ventx2 protein localizes to centrosomes during mitosis and is asymmetrically distribute to one daughter cell. They conclude that in order for embryonic cells to transition from pluripotency to lineage specification, MEK down regulates Ventx2, which would otherwise maintain the pluripotent state much like *Nanog* does in mammals.

The data supporting the conclusion that MEK is required for lineage specification in part through Ventx2 phosphorylation and degradation is strong, but not entirely novel (see below). The main weakness is with the conclusion that MEK inhibition and Ventx2 maintain pluripotency in *Xenopus*, which is not very well supported. Although previous reports suggest that Ventx2 might promote pluripotency, this data is not very robust and is built largely on the fact that Ventx2 can repress differentiation based on a limited marker analysis. Many studies over the past 20 years have shown that MEK and Ventx are involved in mesoderm and neuro-ectoderm development (e.g. PMIDs: 7789277, 7541116, 17525737). In the case of MEK this was attributed to the inducing role of FGF, which has not been ruled out here. Given these previous studies the authors need more rigorous, genome wide analysis, showing that what they are observing is not just the known role of MEK and Ventx in specific lineages. A good test of their hypothesis would be to deplete MEK from animal caps and/or embryos and look at ability of BMP, Activin or Wnt to induce any lineages (including endoderm) by RNA-seq or microarray. Then use a more comprehensive markers analysis (in situ or RT-PCR of a broader panel) to test whether combined MEK and Ventx2 depletion restored all the lineages. Without a genome wide analysis, or some functional assay, showing that the cells cannot differentiate into any lineages it is impossible to conclude that the pluripotency to commitment transition is really involved.

In addition it has already been shown that Ventx2 degradation is promoted GSK3 phosphorylation (PMID:12408807). Since GSK3 has a preference for residues that are primed by MAPK phosphorylation the observation that MEK is required for Ventx2 phosphorylation and degradation is not surprising. In fact the authors should check whether the MEK regulated activity they observe is GSK3-dependent. It is known that Smad1 is phosphorylated by MEK and GSK3, and recruited to the centrosome where it is asymmetrically localized and degraded (PMID:18045539), similar to what is observed here.

Given the known role of MEK (and GSK-3) as well as *Nanog* in mammalian pluripotency, the model that the authors propose is very attractive. If they could provide more rigorous data to support their claim then this would be an important advance.

[Editors’ note: what now follows is the decision letter after the authors submitted for further consideration.]

Thank you for resubmitting your work entitled "Lineage commitment of embryonic cells involves MEK1-dependent clearance of pluripotency regulator Ventx2" for further consideration at *eLife*. Your revised article has been evaluated by Marianne Bronner (Senior Editor), a Reviewing Editor, and one reviewer.

The manuscript has been improved but there are some remaining issues that need to be addressed before acceptance, as outlined below. I would like to stress that *eLife* does not normally allow for multiple rounds of review so it is essential that you complete these relatively minor but important revisions or the paper will not be considered further. However, we will understand if you would prefer to withdraw the paper in the event you are not prepared to perform this essential additional work.

1) RNA encoding the wild type form of MEK1 rescues the MEK1 morpholino phenotype. Seems like a straightforward and necessary experiment that would justify generating the construct.

2) Since it could be done with simple qRT-PCR, it seems important to know whether the Ventx2 morpholino influences the expression of any of the other Ventx genes?

3) Rather than argue the point, the authors should repeat the study on GSK3-MEK interaction as previously suggested ("In fact the authors should check whether the MEK regulated activity they observe is GSK3-dependent."), as it would both solidify previous conclusions and help define the mechanism reported here.

4) As previously suggested, please test the hypothesis by depleting MEK from animal caps and/or embryos and look at ability of BMP, Activin or Wnt to induce any lineages), would address my concerns about whole embryos versus animal caps. In this case, qRT-PCR of a few diagnostic markers (rather than RNA SEQ or microarray analyses) should be sufficient.

---

## [Author Response]

[Editors’ note: the author responses to the first round of peer review follow.]

Although we agree that your work is of potential interest, further experimentation is necessary to convince them of the validity of your conclusions. Among other studies, you would need to use animal caps instead of whole embryos, carry out a more comprehensive analysis to definitely rule out roles for other signaling pathways, perform other types of confirmatory experiments that don't rely on the use of morpholinos and address the problem of maternally-contributed MEK1 as a confounding variable.

Reviewer #1:

This manuscript uses the intact Xenopus laevis embryo as a model system to elucidate the molecular mechanisms involved in the regulation of pluripotency, focussing on MEK1 (map2k1) regulation of Ventx2.2 (encoded by one of a cluster of six related Ventx genes in X. laevis). Ventx proteins, absent in the mouse but present in human, appear to be functionally analogous to the transcription factor Nanog (absent in X. laevis).

The authors demonstrate that MEK1 activity acts to modulate exogenous (and presumably endogenous) Ventx2.2 protein stability; in response to morpholino-mediated down-regulation of MEK1, Ventx2.2 levels increase, the post-mitotic asymmetry in nuclear (exogenous) Ventx2.2 protein disappears, and cellular differentiation is suppressed. The cell differentiation phenotype is rescued in MEK1 morphant embryos by the co-injection of a translation-blocking anti-Ventx2 morpholino.

Experimental design question: While the authors recognize that Xenopus ectodermal explants (animal caps) are effectively pluripotent they have chosen to carry out their studies in whole embryos, which are subject to the complexities of inductive interactions and morphogenetic processes.

Perhaps they could explain their decision, and comment on whether they have examined differentiation in the simpler animal cap system.

Cells of the animal hemisphere of the *Xenopus* blastula embryo are indeed pluripotent, and keep this property when explanted. However, while animal caps have allowed to reveal the inductive capacities of many signalling pathways and transcription factors, they do not accurately reflect in vivo developmental potential (e.g. PMID: 15590738). In particular, injury due to dissection is known to cause ERK phosphorylation, which could complicate our interpretations. Inconsistencies have also been reported in multiple occasions when comparing mouse ES cells to mouse pre-implantation embryos. Thus, we designed our assays to study fate commitment of pluripotent cells in vivo, which allowed us to draw clear conclusions. 1) MEK1 knockdown prevents natural embryonic inductions, and causes the maintenance of high levels of pluripotency genes Ventx2 and Pou5f3.2. The comparison of small Mk-MO injected clones to wild-type surrounding cells in embryos that underwent normal morphogenesis addresses one of the concerns of our reviewer (see Figure 2—figure supplement 2). We also would like to stress that pluripotency gene up-regulation upon MEK1 knockdown in animal caps was reported by RT-qPCR (Figure 2—figure supplement 2). 2) In intact embryos, MEK1-deficient animal cells do not respond to exogenous inducers (Figure 2), but differentiation is restored when Ventx2 is concomitantly knocked down (Figure 4). At best, we can anticipate the same information to be collected from in vitro animal cap assays. 3) Using intact embryos, we were able to analyse sub-cellular Ventx2 protein distribution in animal cells that maintained a natural tissue polarity, which is known to be lost in animal caps.

Reviewer #2:

This work explores the role of MEK and the transcription factor Ventx2 (similar to mammalian Nanog) in regulating the transition from pluripotency to lineage commitment in Xenopus. The authors show that MEK knockdown results in a failure of mesoderm and ectoderm development, based on marker gene analysis. The authors further provide evidence that MEK acts by negatively regulating the mRNA levels and protein stability of Ventx2. They report that Ventx2 protein localizes to centrosomes during mitosis and is asymmetrically distribute to one daughter cell. They conclude that in order for embryonic cells to transition from pluripotency to lineage specification, MEK down regulates Ventx2, which would otherwise maintain the pluripotent state much like Nanog does in mammals.

We did not mention a centrosomal localization of Ventx2 protein during mitosis. We mentioned that: "…Ventx2-Myc was no longer associated with metaphasic chromosomes but rather with mitotic spindles."

The data supporting the conclusion that MEK is required for lineage specification in part through Ventx2 phosphorylation and degradation is strong, but not entirely novel (see below). The main weakness is with the conclusion that MEK inhibition and Ventx2 maintain pluripotency in Xenopus, which is not very well supported. Although previous reports suggest that Ventx2 might promote pluripotency, this data is not very robust and is built largely on the fact that Ventx2 can repress differentiation based on a limited marker analysis.

Pluripotency regulators are indeed characterized by their capacity to refrain differentiation. Ventx2 matches this condition, based on several independent reports, and is used by *Xenopus* experts as a marker of the pluripotent cell state (e.g. PMID: 24210613; 25931449). An unbiased screen identified Ventx as a positive regulator of pluripotency in human ESCs (PMID: 20953172). In zebrafish, a Ventx ortholog known as Vox can reprogram embryonic endodermal cells to a pluripotent state (PMID: 23364327). Thus, the link between Ventx genes and pluripotency is based on robust data in multiple models. Likewise, MEK1 is a known inhibitor of pluripotency in ES cells and our data further support this idea. Finally, Oct4 genes are widely accepted as universal markers of pluripotency in vertebrates and we report here that *Xenopus* Pou5f3.2 (one of several Oct4 orthologs) is repressed by MEK1 through Ventx2.

Many studies over the past 20 years have shown that MEK and Ventx are involved in mesoderm and neuro-ectoderm development (e.g. PMIDs: 7789277, 7541116, 17525737). In the case of MEK this was attributed to the inducing role of FGF, which has not been ruled out here.

MEK1 is indeed required for inductions by FGF, and it is not our intention to rule out this function. What we instead suggest is that when a pluripotent cell is presented with FGF or other RTK ligands, it is able to exit pluripotency and commit to various lineages, depending on the cues it is exposed to. Thus, induction depends on pluripotency exit and these two features are difficult to uncouple. Our work represents a substantial progress in showing that MEK1 knockdown causes the maintenance of two core pluripotency regulators, Ventx2 and *Pou5f3.2*, one of which represents a functional block to differentiation.

Given these previous studies the authors need more rigorous, genome wide analysis, showing that what they are observing is not just the known role of MEK and Ventx in specific lineages. A good test of their hypothesis would be to deplete MEK from animal caps and/or embryos and look at ability of BMP, Activin or Wnt to induce any lineages (including endoderm) by RNA-seq or microarray. Then use a more comprehensive markers analysis (in situ or RT-PCR of a broader panel) to test whether combined MEK and Ventx2 depletion restored all the lineages. Without a genome wide analysis, or some functional assay, showing that the cells cannot differentiate into any lineages it is impossible to conclude that the pluripotency to commitment transition is really involved.

Our reviewer suggests to use genome-wide analysis to evaluate lineage induction in MEK1 and MEK1/Ventx2 morphant situations. We feel that this is overshooting. We used few but extremely well characterized early lineage markers to reveal that MEK1 morphant cells become incompetent to respond to inducers of epidermal, neural, mesodermal and endodermal lineages, and regain this capacity when Ventx2 is also inhibited. What matters here is the competence or incompetence of embryonic cells to respond to inducers. This conclusion will be unchanged if genome-wide analysis is applied. What may be revealed by genome-wide analysis is the identity of putative additional pluripotency regulators; however, this is beyond the scope of our study, which focuses on the specific relationship between MEK1 and Ventx2.

In addition it has already been shown that Ventx2 degradation is promoted GSK3 phosphorylation (PMID:12408807). Since GSK3 has a preference for residues that are primed by MAPK phosphorylation the observation that MEK is required for Ventx2 phosphorylation and degradation is not surprising. In fact the authors should check whether the MEK regulated activity they observe is GSK3-dependent. It is known that Smad1 is phosphorylated by MEK and GSK3, and recruited to the centrosome where it is asymmetrically localized and degraded (PMID:18045539), similar to what is observed here.

The study that reported Ventx2 (also called Xom) degradation in gastrula embryos actually indicated that GSK3 activity is *not* required for degradation (PMID: 12408807 p561). Thus, the degradation mechanism induced by ERK/GSK3 sequential phosphorylation does not apply to Ventx2.

Given the known role of MEK (and GSK-3) as well as Nanog in mammalian pluripotency, the model that the authors propose is very attractive. If they could provide more rigorous data to support their claim then this would be an important advance.

We thank our reviewer for his/her appreciation of our proposed model. We feel, however, that our data are rigorous, notwithstanding the lack of genome-wide analysis.

[Editors’ note: the author responses to the re-review follow.]

The manuscript has been improved but there are some remaining issues that need to be addressed before acceptance, as outlined below. I would like to stress that eLife does not normally allow for multiple rounds of review so it is essential that you complete these relatively minor but important revisions or the paper will not be considered further. However, we will understand if you would prefer to withdraw the paper in the event you are not prepared to perform this essential additional work.

1) RNA encoding the wild type form of MEK1 rescues the MEK1 morpholino phenotype. Seems like a straightforward and necessary experiment that would justify generating the construct.

To address this request, we injected synthetic mRNA encoding a wild-type version of hamster MEK1 ORF into Mk morphant embryos. There was no overlap between Mk-MO and hamster MEK1 mRNA. To further avoid in vitro non-specific interaction between these two molecules, they were injected separately. Wildtype MEK1 could largely restore gastrulation, as well as axial mesoderm and neurectoderm specification. These results now replace those initially obtained with a constitutively active version of MEK1 (Figure 1 and Figure 1—figure supplement 1), and the main text has been modified accordingly (subsection “MEK1 is required for cell competence to differentiate”).

2) Since it could be done with simple qRT-PCR, it seems important to know whether the Ventx2 morpholino influences the expression of any of the other Ventx genes?

We have performed this analysis, and present the corresponding results in Figure 4—figure supplement 1. By RT-qPCR, we observed that *ventx1* and *ventx3*, as well as *pou5f3.1* and *pou5f3.2* expression is lower in Ventx2 morphant embryos at gastrula stage, compared to uninjected control embryos. Conversely, the pro-differentiation marker *gsc*, which is a known negative target of Ventx2 (Sander V et al., 2007; Trindade M et al., 1999), was up-regulated. The same conclusion was obtained through whole-mount in situ hybridization of *ventx1* and *gsc*. We conclude that Ventx2 knockdown leads to a global decrease of Ventx gene expression, and probably activity. This new information is now mentioned in the main text (subsection “Ventx2 inhibition by MEK1 is required for embryonic cell commitment”).

3) Rather than argue the point, the authors should repeat the study on GSK3-MEK interaction as previously suggested ("In fact the authors should check whether the MEK regulated activity they observe is GSK3-dependent."), as it would both solidify previous conclusions and help define the mechanism reported here.

To address whether GSK3 is involved in the MEK1-dependent phenomenon that we have uncovered, we used a dominant-negative version of GSK3, which has been very well characterized in *Xenopus*. The rationale is that if GSK3 and MEK1 collaborate to control pluripotency in *Xenopus* embryos, dnGSK3 should cause similar effects as Mk-MO. First, we verified in a classical assay that dn-GSK3 RNA injection could efficiently induce secondary body axis formation. As expected, RT-qPCR analysis revealed that dn-GSK3 strongly up-regulated the expression of the organizer genes siamois and *gsc* at the early gastrula stage. In contrast, dnGSK3 significantly down-regulated the expression of all *ventx* and *pou5f3* genes in the same embryos. Whole-mount in situ hybridization confirmed that *ventx2* and *pou5f3.2* were down-regulated, whereas *gsc* was up-regulated in embryos injected with dn-GSK3. Thus, MEK1 and GSK3 knockdown had opposite effects on the level of expression of pluripotency regulators. Finally, immunofluorescence analysis revealed that dn-GSK3 did not induce Ventx2-Myc protein stabilization, unlike Mk-MO. This observation is consistent with the data of Zhu et al., (2002), which indicated that GSK3 was not required for Ventx2 proteolysis. Furthermore, Ventx proteins were not recovered in genome-wide GSK3-dependent proteomes (see GSK3 PROTEOME TABLE1 and TABLE2 in http://www.hhmi.ucla.edu/derobertis/; or Acebron S et al., 2014 NCBI GEO accession numbers GSE50629 and GSE50248). We conclude that GSK3 is likely not involved in the transition from pluripotency to commitment of embryonic cells, which primarily involves MEK1-dependent destabilization of the pluripotency regulatory network. These results are presented in the new [Supplementary-material SD3-data] and mentioned in the Discussion section of the main text (second paragraph).

4) As previously suggested, please test the hypothesis by depleting MEK from animal caps and/or embryos and look at ability of BMP, Activin or Wnt to induce any lineages), would address my concerns about whole embryos versus animal caps. In this case, qRT-PCR of a few diagnostic markers (rather than RNA SEQ or microarray analyses) should be sufficient.

As requested, we tested whether animal caps responded to inducers of differentiation in absence of MEK1 activity. We used BMP4, NOGGIN, a low dose of NODAL, and a high dose of NODAL recombinant proteins to induce epidermal, neural, mesodermal and endodermal fates, respectively. We did not use Wnt, as it is known to be a rather poor germ-layer inducer in animal caps. Similar to our initial observations in whole embryos, none of the inducers tested could activate differentiation markers in Mk morphant animal caps, as revealed by RT-qPCR analysis. This analysis confirmed our initial conclusion that MEK1 activity is required for embryonic cell competence to respond to inducers of differentiation. These new results have been included in Figure 2, and are described in the main text (subsection “MEK1 is required for cell competence to differentiate”).